# Co-integrate Col3m *bla*NDM-1-harboring plasmids in clinical *Providencia rettgeri* isolates from Argentina

Denise De Belder,[1,2] Florencia Martino,[1] Nathalie Tijet,[3] Roberto G. Melano,[3,4] Diego Faccone,[1,2] Juan Manuel De Mendieta,[1] Melina Rapoport,[1] Ezequiel Albornoz,[1] Alejandro Petroni,[1] Ezequiel Tuduri,[1] Laura Derdoy,[5] Sandra Cogut,[6] Laura Errecalde,[6] Fernando Pasteran,[1] Alejandra Corso,[1] Sonia A. Gomez[1,2]

**ABSTRACT**   The first cases of *bla*NDM in Argentina were detected in three *Providencia rettgeri* (Pre) recovered from two hospitals in Buenos Aires city in 2013. The isolates were genetically related, but the plasmid profile was different. Here, we characterized the *bla*NDM-1-harboring plasmids of the first three cases detected in Argentina. Hybrid assembly obtained from short- and long-read sequencing rendered *bla*NDM-1 in Col3M plasmids of *ca*. 320 kb (p15268A_320) in isolate PreM15268, 210 kb (p15758B_210) in PreM15758, and 225 kb (p15973A_225) in PreM15973. In addition, PreM15758 harbored a 98-kb circular plasmid (p15758C_98) flanked by a putative recombination site (*hin*-Tn*As2*), with 100% nucleotide ID and coverage with p15628A_320. Analysis of PFGE/S1-nuclease gel, Southern hybridization with *bla*NDM-1 probe, hybrid assembly of short and long reads suggests that pM15758C_98 can integrate by homologous recombination. The three *bla*NDM-1-plasmids were non-conjugative *in vitro*. Moreover, *tra* genes were incomplete, and oriT was not found in the three *bla*NDM-1-plasmids. In two isolates, blaNDM-1 was embedded in a partially conserved structure flanked by two IS*Kox2*. In addition, all plasmids harbored *aph(3')-Ia*, *aph(3')-VI*, and *qnr*D1 genes and *aac(6´)Ib-cr*, *bla*OXA-1, *cat*B3, and *arr3* as part of a class 1 integron. Also, p15268A_320 and p15973A_225 harbored *bla*PER-2. To the best of our knowledge, this is the first report of clinical *P. rettgeri* harboring blaNDM-1 in an atypical genetic environment and located in unusual chimeric Col3M plasmids. The study and continuous surveillance of these pathogens are crucial to tracking the evolution of these resistant plasmids and finding solutions to tackle their dissemination.

**IMPORTANCE**   Infections caused by carbapenem hydrolyzing enzymes like NDM (New Delhi metallo-beta-lactamase) represent a serious problem worldwide because they restrict available treatment options and increase morbidity and mortality, and treatment failure prolongs hospital stays. The first three cases of NDM in Argentina were caused by genetically related *P. rettgeri* recovered in two hospitals. In this work, we studied the genetic structure of the plasmids encoding *bla*NDM in those index cases and revealed the enormous plasticity of these genetic elements. In particular, we found a small plasmid that was also found inserted in the larger plasmids by homologous recombination as a co-integrate element. We also found that the *bla*NDM plasmids were not able to transfer or move to other hosts, suggesting their role as reservoir elements for the acquisition of resistance genes. It is necessary to unravel the dissemination strategies and the evolution of these resistant plasmids to find solutions to tackle their spread.

**KEYWORDS**   NDM-1, *Providencia rettgeri*, plasmid, Tn*As2*, IS*Kox2*

Address correspondence to Sonia A. Gomez, sgomez@anlis.gob.ar.

The authors declare no conflict of interest.

See the funding table on p. 8.

The genus *Providencia* belongs to the order Enterobacterales, family *Morganellaceae*, and includes nine validly named species (*Providencia alcalifaciens*, *P.*

*burhodogranariea*, *P. heimbachae*, *P. rettgeri*, *P. rustigianii*, *P. sneebia*, *P. stuartii*, *P. thailandensis*, and *P. vermicola*) (1). *P. rettgeri* (Pre) is an opportunistic pathogen associated with diverse human infections, such as diarrhea, meningitis, eye infections, catheter-related urinary tract infections, and bacteremia in both hospital and community settings (1). This pathogen is intrinsically resistant to many antimicrobials like ampicillin, first-generation cephalosporins, polymyxins, and tigecycline (2). Furthermore, in recent years, *P. rettgeri* has become increasingly important due to the emergence of carbapenemase-producing strains (2). $bla_{NDM}$ gene is usually carried by transferable plasmids within and between species and has been identified located in plasmids of different incompatibility groups of varying size and resistance determinants (3). Most $bla_{NDM}$-carrying plasmids belong to IncA/C, IncF, IncL/M, IncHI1B, or more recently IncX3 replicon types (4). Moreover, some of these plasmids also harbor other resistance determinants encoding different β-lactamases ($bla_{CMY-2}$ and $bla_{CTX-M-15}$), quinolone resistance (*qnr*B), and 16S rRNA methylases which confer resistance to aminoglycosides (*rmt*C) (5, 6).

$bla_{NDM-1}$ has been widely disseminated to all continents since the first report in 2008 (7, 8). In South America, NDM was reported in Colombia, Ecuador, Brazil, Uruguay, and Argentina in Enterobacterales (9). The plasmids responsible for this dissemination differed between countries: in Colombia and Ecuador, $bla_{NDM-1}$ was located in IncC plasmids, in Brazil in IncFII and IncX3 plasmids (9), and in Uruguay in IncC, IncHI1, and IncM1 (10).

In Argentina, $bla_{NDM-1}$ was first described in 2013 in a Buenos Aires City hospital, from two genetically related *P. rettgeri* isolated from long-term hospitalized patients (11). Both isolates were genetically related, resistant to most antimicrobials and harbored two and four non-transferable plasmids, respectively (11). Three months after the second case, a second institution in Buenos Aires City referred a PER-2-producing *P. rettgeri*, belonging to the same clonal type as the index cases but without an epidemiological link (11). Additionally, Argentina reported $bla_{NDM-1}$ in IncC plasmids in a *Citrobacter amalonaticus*, which also carried *mcr-1*, and the case of a patient harboring five $bla_{NDM-1}$-producing Enterobacterales (12). The aim of the present work was to describe the molecular structure of $bla_{NDM-1}$-harboring plasmids of the first three *P. rettgeri* index cases in Argentina.

## MATERIALS AND METHODS

### Clinical isolates

PreM15268 and PreM15758 were isolated 81 days apart from each other and were partially characterized at the National and Regional Reference Laboratory in Antimicrobial Resistance (NRRLAR) in 2013 (11). A third isolate, PreM15793, was submitted for phenotypic and molecular characterization to the NRRLAR, 3 months after the index cases. Species identification of Pre was done by matrix-assisted laser desorption ionization-time of flight (MALDI-TOF) mass spectrometry (Bruker, Germany). Not*I* pulsed-field gel electrophoresis (PFGE) was performed to determine the genetic relationship. Susceptibility testing was done by agar dilution and automated Phoenix (BD). Metallo-β-lactamase (MBL) production was suspected in isolates that exhibited decreased susceptibility to carbapenems according to the Clinical and Laboratory Standards Institute (CLSI) guidelines (13) and a positive synergism between EDTA and carbapenem disks. PCR and Sanger sequencing (ABI PRISM 3100 o 3730, Applied Biosystems) were used to confirm carbapenemase and ESBL gene variants using the primers listed in Table S1. Biparental conjugation was performed using sodium azide-resistant *Escherichia coli* J53, selecting with several antimicrobials (25 µg/mL ampicillin or 0.5 µg/mL meropenem and 200 µL/mL of sodium azide).

### Short- and long-read sequencing

Genomic DNA was extracted using QIAamp DNA Mini Kit (Qiagen) according to the manufacturer's instructions and eluted in 100 µL of AE buffer. DNA concentration was

determined by Qubit 2.0 Fluorometer (Thermo-Fisher Scientific), and DNA samples were stored at −20°C until further processing. The sequencing library was prepared with the Illumina Nextera XT DNA Library Prep Kit (Illumina, San Diego, CA) as per the manufacturer's instructions. Agilent 2100 Bioanalyzer was used to determine the quality of the DNA library. Sequencing was performed using an Illumina MiSeq platform with 600-cycle MiSeq Reagent Kit v3. Reads were assembled with SPAdes v.3.9.0. Nanopore sequencing was performed on Oxford Nanopore Technologies (ONTs) MinION device with chemistry 8 and flow cells FLO-MIN106 version R9.4. DNA extraction was made with the MasterPure Complete DNA and RNA Purification Kit (Epicenter Illumina), and elution was carried out to a final volume of 40 μL in TE buffer. Libraries for 12 isolates were prepared with the Rapid Barcoding Kit SQK-RBK004 starting with 400 ng of high-molecular-weight DNA from each isolate and according to Oxford Nanopore protocol (RBK_9054_V2_revE_23jan2018). Libraries were loaded and run for 48 hours. Basecalling was performed while sequencing or using Guppy. Nanoplot was used for quality control. Porechop (https://github.com/rrwick/Porechop) was used to split files and trim barcodes. Illumina-ONT hybrid assemblies were performed with Unicycler v0.4.7 (14).

## Bioinformatic analysis of assemblies

The resulting sequences were annotated using Prokka v1.12 (15), and manual curation of the automated annotation was done with Artemis (16). AMRfinderPlus and Plasmid-Finder were used to identify resistance genes and plasmid incompatibility groups, respectively (https://www.genomicepidemiology.org/). Mobile elements and $bla_{NDM-1}$ immediate environment were analyzed manually *in silico* and with PATRIC v 3.6.3, BLASTn, and ISfinder. The origin of transfer site (OriT), relaxase gene, gene encoding type IV coupling protein (T4CP), and gene cluster for bacterial type IV secretion system (T4SS) was searched with OriTFinder. Multiple plasmids analysis and comparison were generated by the BLAST Ring Image Generator (BRIG) v0.95, Geneious Prime v2021.1, and Adobe Illustrator CC v22.0.0. The genetic relationship between isolates was analyzed by calculating a pairwise single nucleotide polymorphism (SNP) distance matrix, built from a FASTA alignment using snp-dists (version 0.6.3, https://github.com/tseemann/snp-dists). The average nucleotide identity between isolates was calculated using an online tool (https://www.ezbiocloud.net) (17). Ribosomal multilocus sequence typing was used to determine the variation of the 53 genes encoding the bacterial ribosome protein subunits (*rps* genes) to integrate microbial taxonomy and typing (18).

To explore the nature of p15758C_98, we used several approaches. We aligned and analyzed the coverage and identity of p15758C_98 among the other plasmids. The fasta file of p15758C_98 was blasted against public databases to find similar or identical elements either alone or integrated (BLASTn last search 28 November 2022). We searched for putative homologous recombination sites manually and with tools such as Tandem Repeats Finder Program (version 4.09), ISfinder, and Artemis. Finally, we mapped the long reads of PreM15758 against the *in silico* built fasta sequence of p15758C_98 and p15758B_210 (p15758D_308) using minimap2 (version 2.17-r941). SAM and BAM files were generated using Samtools (version 1.7 using htslib 1.7–2), and the mapping was visualized with Artemis and Artemis Comparison Tool (version 18.0.3). In addition, the PreM15758 long reads were also mapped against p15628A_320.

## RESULTS

### Epidemiological and phenotypic analysis of PreM15793

PreM15793 was recovered from a screening sample of a patient suffering from ischemic stroke and multiorgan failure in October 2013, 3 months after the second $bla_{NDM-1}$ case (PreM15758) but from a different hospital in Buenos Aires City, with no history of admission to the index cases' institution (Table S2). PreM15793 was identified as *P. rettgeri* (MALDI-TOF score 2.44) and shared the same phenotypic profile as PreM15268

and PreM15758, the index cases (Table S2) (11). They were resistant to all β-lactams, including carbapenems and aztreonam (except for PreM15758), and susceptible to aminoglycosides. $bla_{NDM-1}$ and $bla_{PER-2}$ were detected in PreM15793. Moreover, PFGE revealed that this isolate was genetically related to PreM15268 and PreM15758 (one band of difference in the macro-restriction pattern; Fig. S1). After several attempts, $bla_{NDM-1}$ was not transferred by biparental conjugation, as we had previously reported for the index cases.

## Sequence analysis of $bla_{NDM-1}$-harboring plasmids

The genetic characteristics of each isolate, obtained from hybrid sequences, are shown in Table 1. The reads were assembled into one large chromosome (4.4 Mb) and two to five plasmids (Table 1).

The plasmids obtained by sequencing were consistent in size with the S1 nuclease results for PreM15793 as well as for the index cases with particular findings explained below (Table 1; Fig. 1).

Assessment of the percentage of nucleotide identity between the three plasmids harboring $bla_{NDM-1}$ showed high similarity between them: p15793A_225 and p15758A_210 showed 100% nucleotide identity and 94% coverage, while these plasmids shared 100% identity with 71% and 68% coverage with p15628A_320, respectively (Fig.

**TABLE 1** Plasmid profile and whole genome sequencing (WGS) outcome of $bla_{NDM-1}$-harboring plasmids in clinical *P. rettgeri* isolates, including S1-PFGE results and resistance gene content[c]

| Isolate | Chromosome | Plasmid | Estimated size (kb) (S1-PFGE) | Size (bp) (WGS) | %GC | CDS | Inc-type (WGS) | AMR genes |
|---|---|---|---|---|---|---|---|---|
| PreM15628 | chr[d] | – | – | 4,368,530 | 40.81 | 4581 | – | *dfrA1* |
| | | p15628A_320 | **320** | 319,669 | 45.38 | 385 | Col3M | *arr-3, aac(6')-Ib-cr5, aph(3")-Ib, aph(3')-Ia, aph(3')-VI, aph(6)-Id,* **$bla_{NDM-1}$,** *$bla_{OXA-1}$, $bla_{PER-2}$, catB3, floR, mph(E), msr(E), qnrD1, sul1, sul2, tet(A)* |
| | | p15628B_125 | 142 | 125,717 | 43.52 | 149 | Not typable | |
| PreM15758 | chr | – | – | 4,368,564 | 40.81 | 4568 | – | *dfrA1* |
| | | p15758A_312[a] | **312** | – | – | – | – | – |
| | | p15758B_210 | **210** | 209,862 | 42.25 | 253 | Col3M | *arr-3, aac(6')-Ib-cr5, aph(3')-Ia, aph(3')-VI,* **$bla_{NDM-1}$,** *$bla_{OXA-1}$, catB3, qnrD1, sul1* |
| | | p15758C_98[b] | –[b] | 97,908[b] | 52.22 | 116 | Not typable | *aph(3")-Ib, aph(6)-Id, floR, mph(E), msr(E), sul2, tet(A)* |
| | | p15758D_70 | 79 | 69,881 | 43.14 | 72 | Not typable | |
| | | p15758E_55 | 62 | 55,812 | 44.01 | 81 | Not typable | |
| PreM15793 | chr | – | – | 4,367,913 | 40.47 | 4492 | – | *dfrA1* |
| | | p15793A_225 | **225**[a] | 224,540 | 42.33 | 272 | Col3M | *arr-3, aac(6')-Ib-cr5, aph(3')-Ia, aph(3')-VI,* **$bla_{NDM-1}$,** *$bla_{OXA-1}$, $bla_{PER-2}$, catB3, qnrD1, sul1* |
| | | p15793B_69 | 79 | 69,153 | 43.42 | 79 | Not typable | |
| | | p15793C_57 | | 57,581 | 34.66 | 92 | Not typable | |

[a]This plasmid was only seen in the S1-PFGE gel and hybridized with $bla_{NDM}$ probe, but it was not found by whole genome sequencing (WGS).
[b]The sequence of this plasmid was obtained by hybrid sequencing, but it was not observed in the S1-PFGE/Southern blot (Fig. S2).
[c]Numbers in bold denote an estimated band size that gave positive signal after hybridization with $bla_{NDM-1}$ probe in S1-PFGE/Southern blot (Fig. S2).
[d] chr, chromosome; –, not applicable.

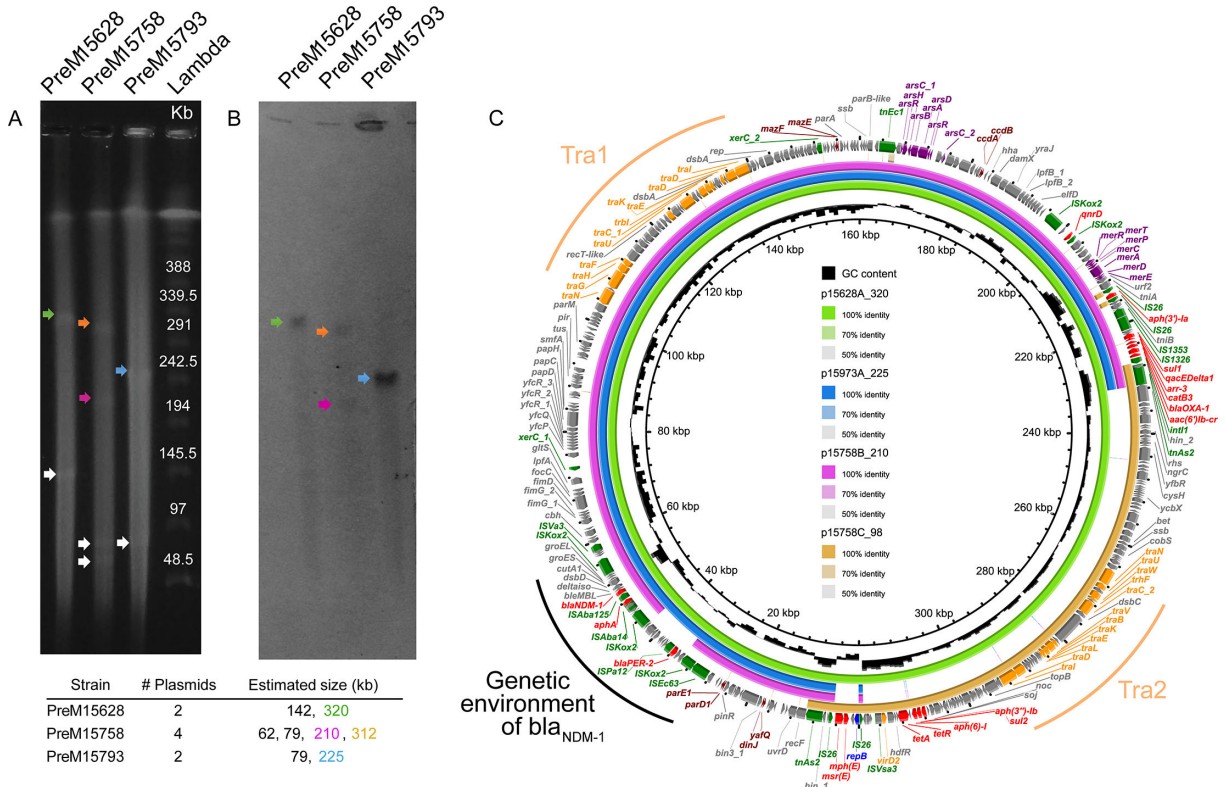

**FIG 1** (A) PFGE of S1-nuclease digested genomic DNA of PreM15628, PreM15758, and PreM15793. The S1-PFGE shows an estimated number of plasmids and sizes (in kilobases), as bands indicated with arrows. (B) Southern blot hybridization membrane with *bla*NDM-1 probe. Black bands on the membrane indicated with colored arrows show positive hybridization signal. The table below summarizes the number of plasmids seen in each isolate and the estimated size by S1-nuclease. Those bands that hybridized with *bla*NDM-1 probe were indicated with the same color code throughout panels A, B, and C. (C) BRIG alignment of *bla*NDM-1-harboring Col3M plasmids. Each plasmid ring was colored according to the bands seen in panels A and B except for p15758C_98, which was not seen in panel A or B, as described in the text. The comparisons were obtained relative to the largest plasmid p15628A_320 (inner ring). At the center of the ring, color levels indicate BLAST result with a matched degree of shared regions. The outermost ring depicts genes as arrows in the corresponding transcription orientation as follows: red, resistance genes; green, mobile genetic elements; orange, transfers genes; brown, toxin/antitoxin system; purple, arsenic and mercury operons; gray, hypothetical proteins. *Tra* modules and *bla*NDM genetic environment are also indicated.

1). Comparison of these three plasmids against the GenBank database revealed no matches over the full length with any reported plasmid. The most closely related was pPrY2001 (accession number NC_022589.1) with 97.9% nucleotide identity and 28%–76% of coverage for the three *bla*NDM-1-harboring plasmids.

The backbone of the *bla*NDM-1-carrying plasmids harbored genes coding for replication (*repB*), plasmid maintenance and stability (ccdA/B, ssb), toxin-antitoxin systems (TA; *mazF/mazE*, ParE/ParD, and YafQ/DinJ), resistance to mercury and arsenic (operons *mer* and *ars*), and conjugative transfer system (Tra1 and Tra2; Fig. 1), but *oriT* was not found. In addition, a 157-bp-long fragment with high identity with Col3M replicon (98.09% identity accession number JX514065) was identified upstream of *qnr*D1 in these three plasmids (19). Interestingly, *qnr*D1 was flanked by two IS*Kox2* in the following order: IS*Kox2*-ORF2-ORF3-ΔORF4-*qnr*D1-ΔIS*Kox2* (Fig. 1). To the best of our knowledge, this *qnr*D1 environment has not been previously reported, and it suggests that *qnr*D1 could be mobilized as part of a composite transposon. Distinctive features of the plasmids were aligned and shown in Fig. 2. The most interesting features were found in the largest plasmid p15628A_320 (~320 kb). Plasmid p15793A_225 was completely contained in p15628A_320, flanked by IS*Kox2* and Tn*As2* (Fig. 2). p15793A_225 harbored *qnr*D1, *aph(3')-Ia,* and a class 1 integron (aligned as an inversion in Fig. 2), and another segment also seen as an inversion with respect to p15628A_320 in Fig. 2, flanked by

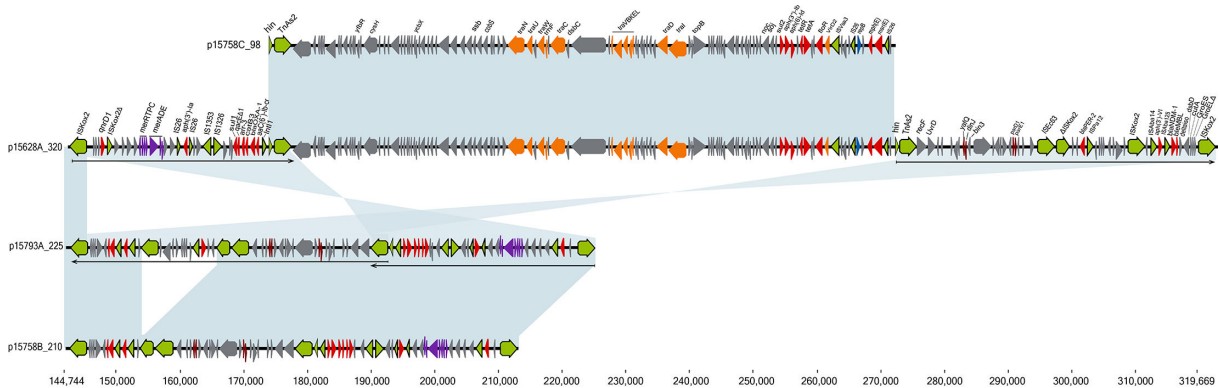

**FIG 2** Partial sequence alignment of the genetic structure of p15628A_320, p15758C_98, p15793A_225, and p15758B_210. The nucleotide sequence of the three longer plasmids was uniformly shortened *in silico* starting in IS*Kox2* and aligned against p15758C_98. Genes are shown as arrows in their transcription orientation. The color code used was: red, resistance genes; green, mobile genetic elements (integrases, insertion sequences, or transposons); orange, plasmid transfer genes; purple, mercury resistance operon; brown, toxin/antitoxin system; gray, hypothetical proteins. Light blue shadow blocks represent 100% nucleotide ID between sequences, and the thin black arrows indicate inversions.

the same transposons (Tn*As2* and IS*Kox2*) harboring $bla_{NDM-1}$ and $bla_{PER-2}$. Of note, $bla_{PER-2}$ was detected in p15628A_320 and p15793A_225 but not in p15758B_210. In addition, this plasmid harbored a ca. 98-kb insertion, identical to p15758C_98 obtained in the second clinical isolate PreM15758 (Table 1; Fig. 1 and 2). When looking at the S1-nuclease/PFGE results obtained for PreM15758 (Table 1; Fig. 1), two bands were seen that hybridized with $bla_{NDM}$ probe of ca. 312 and 210 kb. WGS assembly instead rendered p15758C_98 and p15758B_210 circular sequences. Based on these findings, we hypothesized that p15758C_98 is a genetic element able to be integrated into, or excised from, larger plasmids like p15758B_210 or p15758A_312, respectively, by homologous recombination. As a consequence, p15758A_312, p15758B_210, and p15758C_98 may co-exist in the bacterial clone of Pre15758, at least temporarily. To test this hypothesis, first, we searched for similar elements to p15758C_98 with the Blast search tool. GenBank MZ465529.1 was the only matching sequence, obtained in Argentina from an *E. coli* urine sample, with 99.99% identity and 100% query cover, integrated into an *IncC* plasmid harboring $bla_{OXA-2}$, $bla_{PER-2}$, and *aac(6')-Ib-cr* resistance genes (unpublished). Second, we searched for putative homologous recombination sites at the boundaries of the 98 kb sequence. To do this, we first analyzed the boundaries of the 98 kb element inserted in p15628A_320 (Fig. 2). Interestingly, the 98 kb sequence was flanked by two copies of a DNA-invertase (*hin*; UniProt P03013, last entry 18 January 2023) and the transposon Tn*As2*. Third, we analyzed the plasmids in PreM15758, finding only one copy of *hin*-Tn*As2* in p15758C_98 and p15758B_210 (Fig. 2), suggesting this may act as the homologous recombination site. Finally, we mapped PreM15758 long reads against the *in silico* concatenated fasta sequence inserting p15758C_98 into the fasta sequence of p15758B_210 (p15758_98–210) as in p15628A_320. As a result, we found long reads along the whole sequence of p15758_98–210 with different depth of coverage (Fig. S2A), demonstrating the existence of a smaller population (low copy number) with the three elements in the cell (p98 p210 p312). In addition, there were reads that started strictly at both ends of the 98 kb segment indicating the presence of this element on its own (Fig. S2B). The same results were observed when mapping PreM15758 long reads against p15628A_320. Altogether, these findings support the hypothesis of the co-existence of the three plasmids mediated by the integration or excision of p15758C_98 into p15758B_210 through homologous recombination via *hin*-Tn*As2*.

In all cases, $bla_{NDM-1}$ was embedded in a partially conserved structure flanked by two IS*Kox2* as follows: IS*Kox2* - IS*Aba14* - *aph(3')-VI* –IS*Aba125* – $bla_{NDM-1}$ – $ble_{MBL}$ – Δ*iso* – *tat* – *cutA1* – *gro*ES – Δ*gro*EL – IS*Kox2*. In p15758B_210, the IS*Kox2* downstream from *gro*EL was truncated (Fig. 1 and 2). Moreover, *aac(6')-Ib-cr*, $bla_{OXA-1}$, *catB3*, and *arr-3* were found as

gene cassettes of a classical class 1 integron. The genes *aph(3″)-Ib*, *aph(6)-Id*, *floR*, *mph(E)*, *msr(E)*, *tet(A)*, and *sul2* were found only in p15628A_320 and p15758C_98 (Table 1; Fig. 1).

## DISCUSSION

*P. rettgeri* is an opportunistic pathogen mainly associated with urinary tract infections that has been gaining clinical relevance due to its association with NDM (3). In this work, we completely characterized three *bla*<sub>NDM-1</sub> plasmids harbored in the first *P. rettgeri* clinical isolates reported in Argentina in 2014 (11). *P. rettgeri* producing NDM-1 has been reported in several continents including America (20–23). In South America, *P. rettgeri*-NDM was reported in Brazil where $bla_{NDM-1}$ was located in the chromosome (24), Colombia, in a non-typable plasmid (3, 25), Argentina (11), Ecuador (26), and Uruguay (10). It is known that $bla_{NDM-1}$ is frequently associated with different replicons in Enterobacterales (3). In this report, $bla_{NDM-1}$ was located in an unusual non-conjugative replicon Col3M. This replicon type has been reported as a small non-conjugative plasmid that carries *qnrD* in tribe Proteeae (27). In that report, the genetic environment of *qnrD1* was similar but with different gene order.

As mentioned above, the plasmids described here were non-conjugative or non-mobilizable. This result could in part be supported by the absence of OriT required for conjugation or mobilization (28), although further analysis and experiments are required. Usually, we search for the conjugative or transfer machinery to explain or determine the likelihood for dissemination of the plasmids to other bacterial clones or species. In this case, the isolates studied carried several other non-typable plasmids that may contribute with the mobilization of the resistance plasmids (28). In addition, the $bla_{NDM}$ plasmids described showed mosaic structures made of genetic elements like transposons (*hin*-TnAs2) and insertion sequences (IS*26*) that suggest the acquisition or loss of genes in and out of the plasmid and/or the cell or by homologous recombination (Fig. 2) (29). The gene, *hin*, is described as a site-specific recombinase resolvase family, also known to act as a DNA-invertase (30). Here, we found *hin* next to Tn*As2* transposon twice in p15628A_320 and as a single copy in the rest of the plasmids described here (Fig. 2). In addition, we also observed that *hin*-Tn*As2* was flanking the 98-kb fragment (p15758B_98) when inserted in p15628A_320. Therefore, we hypothesized that *hin*-Tn*As2* could be involved in the insertion of p15758B_98 (98 kb) fragment into a plasmid similar to p15793A_225 that yielded the p15628A_320 plasmid by homologous recombination (Fig. S2). This hypothesis is supported by previous findings where Tn*As2*, a Tn3-family transposon, was associated with *mcr-3* gene and identified as a key element in its transposition (31) and those reporting the role of *hin* in DNA recombination (30, 32). Nevertheless, additional experiments are necessary to prove this hypothesis.

Two IS*Kox2* (also known as Tn*6935*), in the same orientation, either complete or truncated, flanked the immediate genetic environment of $bla_{NDM-1}$, similar to the structure reported by Papa-Ezdra in Uruguay (10). Originally, $bla_{NDM}$ was described inserted in the composite transposon Tn*125* bound by two IS*Aba125*, but soon after, it was reported in a variety of genetic contexts including variants of Tn*125* either truncated or interrupted (33). This interruption was usually due to the insertion of other mobile genetic elements such as IS*1*, IS*5*, IS*26*, IS*903*, IS*Ec33*, IS*Kpn14*, among others (34). Genetic recombination generates a great variety of complex genetic contexts for $bla_{NDM-1}$ close environment. Considering our results and those from Uruguay, it seems that this transposable unit is circulating in our region.

Another interesting finding was the fact that, in these highly similar plasmids, the sequence that contained *bla*<sub>PER-2</sub> was only present in p15628A_320 and p15793A_225 flanked by IS*Kox2* and IS*Pa12*. This same genetic structure was also seen in a previous report from Argentina, where $bla_{PER-2}$ was located in an IncA/C$_1$ plasmid (pCf587). That plasmid was found in a clinical isolate of *Citrobacter freundii* 33587, which was recovered from a urine sample in 1999 (35).

## Conclusion

Here, we described the complete sequences of three unusual Col3M plasmids identified in the first three clinical isolates of *P. rettgeri* harboring $bla_{NDM-1}$ in Argentina. These plasmids were found to be non-conjugative, suggesting their role as reservoir elements for the acquisition of resistance genes. Considering the genetic complexity of these multidrug resistance isolates, further studies are necessary to understand their dissemination strategies.

## ACKNOWLEDGMENTS

We would like to thank Stella Cristaldo for her technical assistance.

This work was funded by the Agencia Nacional de Promoción Científica y Tecnológica (ANPCyT; Préstamo BID, PICT-2017-0321) to S.A.G. and D.F. and by the annual budget of the Ministry of Health to the NRRLAR. The present document was prepared at the NRRLAR, under the project "Working together to combat antimicrobial resistance" led by the Tripartite Alliance FAO-PAHO-WOAH with financial support from the European Union (EU). The contents of this manuscript do not represent the views and opinions of FAO, PAHO, WOAH, or the EU.

The funders were not involved in the study design; in the collection, analysis, and interpretation of data; in the writing of the report; and in the decision to submit the article for publication. The authors declare that the research was conducted in the absence of any commercial or financial relationships that could be construed as a potential conflict of interest.

## AUTHOR AFFILIATIONS

[1]Antimicrobial Agents Division, National and Regional Reference Laboratory in Antimicrobial Resistance (NRRLAR), National Institute of Infectious Diseases–ANLIS "Dr. Carlos G. Malbrán", Buenos Aires, Argentina

[2]National Council on Scientific and Technical Research (CONICET), Buenos Aires, Argentina

[3]Public Health Ontario Laboratory, Toronto, Ontario, Canada

[4]Department of Laboratory Medicine and Pathobiology, University of Toronto, Toronto, Ontario, Canada

[5]Hospital General de Agudos José María Ramos Mejía, Buenos Aires, Argentina

[6]Hospital General de Agudos Dr. Juan A. Fernández, Buenos Aires, Argentina

## AUTHOR ORCIDs

Sonia A. Gomez  http://orcid.org/0000-0001-9106-1762

## FUNDING

| Funder | Grant(s) | Author(s) |
| --- | --- | --- |
| MINCyT | Agencia Nacional de Promoción Científica y Tecnológica (ANPCyT) | PICT-2017-0321 | Sonia A. Gomez |

## AUTHOR CONTRIBUTIONS

Denise De Belder, Conceptualization, Data curation, Formal analysis, Investigation, Methodology, Software, Validation, Visualization, Writing – original draft, Writing – review and editing | Florencia Martino, Data curation, Formal analysis, Investigation, Methodology, Software, Visualization, Writing – review and editing | Nathalie Tijet, Data curation, Formal analysis, Investigation, Methodology, Software, Validation, Visualization, Writing – review and editing | Roberto G. Melano, Conceptualization, Data curation, Formal analysis, Investigation, Methodology, Software, Supervision, Validation, Visualization, Writing – review and editing | Diego Faccone, Funding acquisition,

Project administration, Resources, Supervision, Validation, Writing – review and editing | Juan Manuel De Mendieta, Data curation, Formal analysis, Investigation, Methodology, Software | Melina Rapoport, Data curation, Investigation, Methodology, Validation, Writing – review and editing | Ezequiel Albornoz, Data curation, Investigation, Methodology, Visualization | Alejandro Petroni, Data curation, Formal analysis, Investigation, Methodology, Validation, Visualization, Writing – review and editing | Ezequiel Tuduri, Data curation, Investigation, Methodology, Software, Validation | Laura Derdoy, Data curation, Methodology, Resources, Software, Writing – review and editing | Sandra Cogut, Data curation, Investigation, Methodology, Visualization, Writing – review and editing | Laura Errecalde, Data curation, Investigation, Methodology, Visualization | Fernando Pasteran, Conceptualization, Investigation, Project administration, Supervision, Validation, Writing – review and editing | Alejandra Corso, Conceptualization, Resources, Supervision, Validation, Visualization, Writing – review and editing | Sonia A. Gomez, Conceptualization, Data curation, Formal analysis, Funding acquisition, Investigation, Project administration, Resources, Supervision, Validation, Visualization, Writing – original draft, Writing – review and editing

## DATA AVAILABILITY

Sequence data from this article were deposited with the GenBank Data Libraries under Accession No.: Bioproject PRJNA735191 associated to the Biosamples SAMN19573044 for PreM15793 GenBank CP123366, CP123367, CP123368, SAMN19573043 for PreM15758 GenBank CP123369, CP123370, CP123371 and CP123372, and SAMN19571474 for PreM15628 GenBank CP123373, CP123374, CP123375.

## ADDITIONAL FILES

The following material is available online.

### Supplemental Material

**Supplemental material (Spectrum01651-23-s0001.pdf).** Tables S1 and S2; Fig. S1 and S2.

### Open Peer Review

**PEER REVIEW HISTORY (review-history.pdf).** An accounting of the reviewer comments and feedback.

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
