## [Reviewer comments · Microbiology Spectrum]

Microbiology Spectrum

Co-integrate Col3M bla_{NDM-1}-harbouring plasmids in clinical *Providencia rettgeri* isolates from Argentina

Denise De Belder, Florencia Martino, Nathalie Tijet, Roberto Melano, Diego Faccone, Juan De Mendieta, Melina Rapoport, Ezequiel Albornoz, Alejandro Petroni, Ezequiel Tuduri, Laura Derdoy, Sandra Cogut, Laura Errecalde, Fernando Pasteran, Alejandra Corso, and Sonia Gomez

Corresponding Author(s): Sonia Gomez, Servicio Antimicrobianos, Departamento de Bacteriología, Instituto Nacional de Enfermedades Infecciosas - ANLIS "Dr. Carlos G. Malbrán"

Review Timeline:

Submission Date:	May 2, 2023
Editorial Decision:	June 26, 2023
Revision Received:	July 6, 2023
Accepted:	August 1, 2023

Editor: Rafael Vignoli

Reviewer(s): Disclosure of reviewer identity is with reference to reviewer comments included in decision letter(s). The following individuals involved in review of your submission have agreed to reveal their identity: Milena Droga (Reviewer #2)

Transaction Report:

DOI: <https://doi.org/10.1128/spectrum.01651-23>

June 26, 2023

Dr. Sonia A Gomez

Servicio Antimicrobianos, Departamento de Bacteriología, Instituto Nacional de Enfermedades Infecciosas - ANLIS "Dr. Carlos G. Malbrán"
Ciudad Autónoma de Buenos Aires
Argentina

Re: Spectrum01651-23 (Co-integrate Col3M bla_{NDM-1}-harbouring plasmids in clinical Providencia rettgeri isolates from Argentina)

Dear Dr. Sonia A Gomez:

Thank you for submitting your manuscript to Microbiology Spectrum. Your paper has been reviewed by two experts in the field and they have pointed out some modifications that need to be addressed before accepting your paper. When submitting the revised version of your paper, please provide (1) point-by-point responses to the issues raised by the reviewers as file type "Response to Reviewers," not in your cover letter, and (2) a PDF file that indicates the changes from the original submission (by highlighting or underlining the changes) as file type "Marked Up Manuscript - For Review Only". Please use this link to submit your revised manuscript - we strongly recommend that you submit your paper within the next 60 days or reach out to me. Detailed instructions on submitting your revised paper are below.

Link Not Available

Sincerely,

Rafael Vignoli

Journals Department
Reviewer comments:

Reviewer #1 (Comments for the Author):

The article describes the plasmids carrying NDM-1 from three clinical isolates obtained in Argentina, as well as possible routes of mobilization through transposable genes. It is necessary to improve the quality of the images since they make the review and analysis of the same very difficult.

Mentions are made that cannot be verified, given their quality.

Additional comments, are described below.

Abstract

- finding solutions

Introduction

- and includes
- and bacteremia
- like ampicillin
- first-generation
- Please check reference 6 concerning the mentioned genes.
- has been widely
- an epidemiological link
- ionization-time of
- azide-resistant

2. Materials and Methods 2.1 Clinical isolates

- reference 11 is missing in the text
- table S1 presents epidemiological information and antibiotics MIC not primers list
- per the manufacturer's instructions.
- the quality of the DNA library.
- and elution was carried out
- split files and trim barcodes
- ResFinder and PlasmidFinder
- was analyzed by calculating a pairwise

Results

- phenotypic profile as PreM15268 and
- the text mentioned all isolates were resistant to aztreonam, but in table S1 strain PreM15758 is susceptible.
- The quality of figure 2 is not adequate and makes it impossible to analyze the alignment figure.
- Is not clear the comment concerning p15758A_312, since this plasmid is only mentioned in line 196 and no data is available in table 1.
- integrated into an IncC plasmid
- it is mentioned that all isolates are susceptible to amikacin, even though the three of them harbored aac(6')Ib-cr and aph genes.

Discussion

- known to act as a DNA

Reviewer #2 (Comments for the Author):

The article describes the genome of the first three NDM-1 producing *P. rettgeri* strains isolated in Argentina. In addition to the detailed study of the plasmids that carry the blaNDM-1 gene, a very interesting homologous recombination event is also described in this clone, which results in distinct Col3M plasmids. The genetic environment that surrounds the blaNDM-1 gene is not the most common one found in Latin America, but it was recently described in Uruguay and its dissemination deserves attention. Below are minor corrections:

Line 39: Morganellaceae

Line 87: were, instead of was

Line 97: ...to determine the quality...

Line 114: and and

Line 119: I guess there is a comma instead of a period after v.0.95

Line 129: ...similar or identical...

Line 139: was, instead of were

Lines 189-190: "Of note, blaPER-2 was only found in p15628A_320."

Actually, it was also found in p15793A_225. Maybe you mean that between the index strains, it was only found in p15628A_320. Please clarify the sentence.

Line 253: as as

Staff Comments:

Preparing Revision Guidelines

To submit your modified manuscript, log onto the eJP submission site at <https://spectrum.msubmit.net/cgi-bin/main.plex>. Go to

Author Tasks and click the appropriate manuscript title to begin the revision process. The information that you entered when you first submitted the paper will be displayed. Please update the information as necessary. Here are a few examples of required updates that authors must address:

Please return the manuscript within 60 days; if you cannot complete the modification within this time period, please contact me. If you do not wish to modify the manuscript and prefer to submit it to another journal, please notify me of your decision immediately so that the manuscript may be formally withdrawn from consideration by Microbiology Spectrum.

REVIEWER #1

We would like to thank reviewer #1 for the constructive revision of our manuscript. We have accepted and answered all the queries raised. Our responses can be found below each comment

Reviewer #1 (Comments for the Author):

The article describes the plasmids carrying NDM-1 from three clinical isolates obtained in Argentina, as well as possible routes of mobilization through transposable genes. It is necessary to improve the quality of the images since they make the review and analysis of the same very difficult.

Mentions are made that cannot be verified, given their quality.

Additional comments, are described below.

Abstract

- finding solutions **Accepted:**

Introduction

- and includes **Accepted**

- and bacteremia **Accepted**

- like ampicillin **Accepted**

- first-generation **Accepted**

- Please check reference 6 concerning the mentioned genes. **Accepted. Thank you for your observation. The reference was changed accordingly**

- has been widely **Accepted**

- an epidemiological link **Accepted**

- ionization-time of **Accepted**

- azide-resistant **Accepted**

2. Materials and Methods 2.1 Clinical isolates

- reference 11 is missing in the text **Accepted. The mistake was fixed.**

- table S1 presents epidemiological information and antibiotics MIC not primers list

Accepted, Thank you for the observation. The primer list was added to the supplementary material as Table S1 and the table containing the epidemiological data was re named to Table S2.

- per the manufacturer's instructions. **Accepted**
- the quality of the DNA library. **Accepted**
- and elution was carried out **Accepted**
- split files and trim barcodes **Accepted**
- ResFinder and PlasmidFinder **Accepted**
- was analyzed by calculating a pairwise **Accepted**

Results

- phenotypic profile as PreM15268 and **Accepted**

- the text mentioned all isolates were resistant to aztreonam, but in table S1 strain PreM15758 is susceptible. **Accepted. Thank you for the observation. The sentence was modified accordingly**

- The quality of figure 2 is not adequate and makes it impossible to analyze the alignment figure. **Accepted. Thank you for the observation. I replaced figure 1 and figure 2 by the tif 600 dpi images.**

- Is not clear the comment concerning p15758A_312, since this plasmid is only mentioned in line 196 and no data is available in table 1.
Thank you for your comment. We mentioned p15758A_312 because PreM15758 showed a 312 Kb band seen in the S1-PFGE gel with positive hybridization using the *bla*_{NDM} probe. When it was sequenced, we did not find a 312 Kb plasmid. This is why Table 1 only has the 312 bold number (explained in the table 1 footnote) and no additional information. Moreover, we mention p15758A_312 in Lines 198 -201 to explain / hypothesize the fact that p15758C_98 may be “inserting” or “excising” from the larger plasmid by homologous recombination. We have now modified the table’s footnote and added details to make it clearer and minimize confusion.

- integrated into an IncC plasmid **Accepted.**

- it is mentioned that all isolates are susceptible to amikacin, even though the three of them harbored *aac(6')Ib-cr* and *aph* genes.

Yes, in fact the three isolates are susceptible to gentamicin and amikacin. The susceptibility to certain aminoglycosides in *Providencia* spp harboring these genes has been documented in several articles, although, to the best of our knowledge, the reason has not been explained. In particular, susceptibility to amikacin by *P. rettgeri* strains was reported by Ingo Stock and B. Wiedemann in "Natural antibiotic susceptibility of *Providencia stuartii*, *P. rettgeri*, *P. alcalifaciens* and *P. rustigianii* strains Antimicrobial susceptibility" *J. Med. Microbiol.* - Vol. 47 (1 998), 629-642. In the case of *aac(6')Ib-cr*, this modifying enzyme mostly affects ciprofloxacin and norfloxacin by acetylation, but the inactivation effect over gentamicin and amikacin is either poor or variable. In our experience, *aac(6')Ib-cr* does not affect gentamicin and amikacin, as reported previously (see Table 2 in Andrés, P. *et. al. Antimicrobial Agents and Chemotherapy*, 2013, 57 (6), p. 2467–2475.). In our epidemiology, high level resistance to amikacin and gentamicin is mostly seen when *rmt* enzymes are present inserted in the genetic element with *bla*_{NDM} (usually in the A/C plasmid), or with *aac(6')-Ib*, which confers resistance to gentamicin and amikacin.

REVIEWER #2

We would like to thank reviewer #2 for the constructive revision of our manuscript. We have accepted and answered all the queries raised. Our responses can be found below each comment

Reviewer #2 (Comments for the Author):

The article describes the genome of the first three NDM-1 producing *P. rettgeri* strains isolated in Argentina. In addition to the detailed study of the plasmids that carry the blaNDM-1 gene, a very interesting homologous recombination event is also described in this clone, which results in distinct Col3M plasmids. The genetic environment that surrounds the blaNDM-1 gene is not the most common one found in Latin America, but it was recently described in Uruguay and its dissemination deserves attention. Below are minor corrections:

Line 39: Morganellaceae **Accepted**

Line 87: were, instead of was **Accepted**

Line 97: ...to determine the quality... **Accepted**

Line 114: and and **Accepted**

Line 119: I guess there is a comma instead of a period after v.0.95 **Accepted**

Line 129: ...similar or identical... **Accepted**

Line 139: was, instead of were **Accepted**

Lines 189-190: "Of note, blaPER-2 was only found in p15628A_320."
Actually, it was also found in p15793A_225. Maybe you mean that between the index strains, it was only found in p15628A_320. Please clarify the sentence.

Accepted, the sentence was clarified.

Line 253: as as **Accepted**

August 1, 2023

Dr. Sonia A Gomez

Servicio Antimicrobianos, Departamento de Bacteriología, Instituto Nacional de Enfermedades Infecciosas - ANLIS "Dr. Carlos G. Malbrán"
Ciudad Autónoma de Buenos Aires
Argentina

Re: Spectrum01651-23R1 (Co-integrate Col3M bla_{NDM-1}-harbouring plasmids in clinical Providencia rettgeri isolates from Argentina)

Dear Dr. Sonia A Gomez:

Congratulation!! Your manuscript has been accepted, and I am forwarding it to the ASM Journals Department for publication. You will be notified when your proofs are ready to be viewed.

Sincerely,

Rafael Vignoli
Editor, Microbiology Spectrum
